# Characterization by MALDI-TOF MS and 16S rRNA Gene Sequencing of Aerobic Endospore-Forming Bacteria Isolated from Pharmaceutical Facility in Rio de Janeiro, Brazil

**DOI:** 10.3390/microorganisms12040724

**Published:** 2024-04-03

**Authors:** Nathalia Gonçalves Santos Caldeira, Maria Luiza Soares de Souza, Rebeca Vitória da Silva Lage de Miranda, Luciana Veloso da Costa, Stephen James Forsythe, Viviane Zahner, Marcelo Luiz Lima Brandão

**Affiliations:** 1National Institute for Quality Control in Health, Oswaldo Cruz Foundation, Rio de Janeiro 21040-900, Brazil; nathalia.caldeira@fiocruz.br; 2Integrated Laboratory–Simulids and Onchocerciasis & Medical and Forensic Entomology, Oswaldo Cruz Institute, Oswaldo Cruz Foundation, Rio de Janeiro 21040-900, Brazil; marialuiza937@gmail.com; 3Institute of Technology in Immunobiologicals, Oswaldo Cruz Foundation, Rio de Janeiro 21040-900, Brazil; rebeca.lage@bio.fiocruz.br (R.V.d.S.L.d.M.); marcelo.brandao@bio.fiocruz.br (M.L.L.B.); 4Foodmicrobe.com Ltd., Adams Hill, Keyworth, Nottingham NG12 5GY, UK; sforsythe4j@gmail.com

**Keywords:** *Bacillus*, endospore-forming bacteria, MALDI TOF/MS, 16S rRNA sequencing, pharmaceutical industry, identification, quality control

## Abstract

*Bacillus* and related genera are among the most important contaminants in the pharmaceutical production environment, and the identification of these microorganisms at the species level assists in the investigation of sources of contamination and in preventive and corrective decision making. The aim of this study was to evaluate three methodologies for the characterization of endospore-forming aerobic bacterial strains isolated from a pharmaceutical unit in Rio de Janeiro, Brazil. MALDI-TOF MS was performed using MALDI Biotyper^®^ and VITEK^®^ MS RUO systems, and complete 16S rRNA gene sequencing was performed using the Sanger methodology. The results showed the prevalence of the genera *Bacillus* (n = 9; 36.0%), *Priestia* (n = 5; 20.0%), and *Paenibacillus* (n = 4; 16.0%). Three (20.0%) strains showed <98.7% of DNA sequencing similarity on the EzBioCloud Database, indicating possible new species. In addition, the reclassification of *Bacillus pseudoflexus* to the genus *Priestia* as *Priestia pseudoflexus* sp. nov. is proposed. In conclusion, 16S rRNA and MALDI TOF/MS were not sufficient to identify all strains at the species level, and complementary analyses were necessary.

## 1. Introduction

The presence of microorganisms in pharmaceutical production environments, such as water, raw materials, surfaces, machines, and employees, can represent a risk for product safety [1,2]. To minimize microbial contamination risk for product safety, Good Manufacturing Practices (GMPs) must be followed to ensure that biologicals meet, among other parameters, acceptable limits for microorganisms [3,4].

Quality control methods in laboratories in pharmaceutical industries must be capable of detecting and identifying contaminants from various sources, including production environments. The identification of isolates at the species level and possibly subspecies level can be necessary [5].

Grade A and B areas, for example, are sensitive areas during pharmaceutical production, as they are where in aseptic production, aseptic connections and high-risk operations occur, such as filling; there are also lid reservoirs, open ampoules, and vials [4]. According to European Union (EU) Annex 1, microorganisms isolated from these areas must be identified to the species level, and their impact on product quality must be assessed [3]. Another point to be considered is the type of sample from which the contaminant was isolated. Microorganisms obtained from sterility tests of finished products and aseptic simulations must be identified at least to the species level. This will facilitate both the investigation of possible sources of contamination and the making of preventive and corrective decisions [3,4,5]. The identification of microorganisms detected in areas of grades C and D, which are those where less critical steps are carried out in the manufacture of sterile medicines, such as the preparation of solutions to be filtered and the handling of materials after washing, can reveal organisms that are difficult to control. Such organisms include spore-forming microorganisms and molds [3,4]. Other undesirable environmental contaminants include potential biofilm-producing microorganisms, and/or producers of various degradative enzymes that are not eliminated by filtration. Monitoring for these organisms is important for the quality, efficacy, and stability of the final product [6].

Monitoring the pharmaceutical production environment, beyond establishing limits, must allow for the detection and identification of contaminants from different sources.

On a global scale, data on microorganism strains from the pharmaceutical industry are still scarce in the literature. Previous studies have shown that *Bacillus* and related genera are among the most common contaminants of these environments [6]. The ability to produce spores allows this bacterial group to remain in the environment for long periods due to the resistance of the spores to temperature variations and the sanitizers used in industry [7,8].

The methodology used to identify strains from pharmaceutical production environments must consider the accuracy of the results and the time required to obtain them, as the release of batches of these products depends on this [9]. Matrix-Assisted Laser Desorption Ionization–Time of Flight/Mass Spectrometry (MALDI-TOF MS) technology has proven to be a good option for the identification of microorganisms of pharmaceutical origin. It has several advantages over traditional phenotypic methods, such as speed and higher specificity and sensitivity [6,9]. Despite the advantages presented by MALDI-TOF MS, this methodology is still unable to identify many pharmaceutical production environment microorganisms [6,7]. This method was initially developed to identify mainly strains isolated from clinical specimens [10], which means that the relevant databases do not yet include all microorganisms that can be found in pharmaceutical production environments [6,7].

The molecular methodology of 16S rRNA gene sequencing has also been used for identifying bacterial strains from pharmaceutical production environments [6,7]. However, this methodology has some disadvantages, such as the inability to differentiate among closely related species with practically identical 16S rRNA genes, for example, *Bacillus* and related genera [6,7]. In these cases, the analysis of species-specific genes or whole-genome sequencing (WGS) can be used to achieve identification at the species level [7,11].

In addition to the advantages mentioned above, the rapid analysis of fungal and bacterial isolates and the low cost of reagents make MALDI-TOF MS an affordable option for microbiological quality control in laboratories in pharmaceutical industries. In contrast, genotypic methodologies can have a high cost and require longer analysis time. In cases where isolates are not initially identified due to the limitations of the MALDI-TOF database, this can be remedied by updating of the database after sequencing the 16S rRNA gene [7,10,11].

The aim of this study was to evaluate three methodologies for the characterization of endospore-forming aerobic bacterial strains isolated from a pharmaceutical unit in Rio de Janeiro, Brazil.

## 2. Materials and Methods

### 2.1. Bacterial Strains

Twenty-five strains were isolated from different samples and stages of the production chain of a pharmaceutical unit producing immunobiologicals from 2015 to 2020 (Table 1). They were obtained from sources such as air monitoring, water vapor, operator monitoring, bioburden test, potable water, and sterility test, according to the Brazilian Pharmacopoeia [4].

The strains had been previously characterized as endospore-forming aerobic bacteria after Gram staining and micro- and macroscopic observation and analyzed by VITEK^®^2 Compact System (bioMérieux, Craponne, France) but had not been identified.

Stock cultures of these strains were prepared and maintained at <−70 °C in Difco™ Skim Milk 30% (BD Biosciences, Le Pont de Claix, France) containing 30% glycerol (Merck KGaA, Darmstadt, Germany). For daily use, strains were plated on Tryptic Soy Agar (TSA) plates and incubated at 30–35 °C for 24–72 h.

### 2.2. Identification by MALDI-TOF MS

Proteomic characterization by MALDI-TOF MS was performed using two semi-automated systems: MALDI Biotyper^®^ (MBT) (Bruker Corporation, Billerica, MA, USA) and VITEK^®^ MS RUO (bioMérieux, Craponne, France), according to the manufacturers’ instructions. Both systems record mass spectra from 2000 to 20,000 Da and have 60 and 50 HZ laser frequencies, respectively. MALDI Biotyper^®^ has a confidence interval reported as score values of 0.00–1.69 (no organism identification possible), 1.70–1.99 (low-confidence species identification), and 2.00–3.00 (high-confidence species identification). VITEK^®^ MS RUO has a confidence interval reported value in percentage, and the strains is considered identified when a percentage ≥75.0 is obtained.

For analysis, a small portion of the colony was applied, in duplicate, to a specific slide for each piece of equipment, and after drying, 1 µL of alpha-cyano-4-hydroxycinnamic acid matrix (VITEK MS-CHCA; bioMérieux, Craponne, France; Bruker Matrix HCCA, Bruker Corporation, Billerica, MA, USA) was added to the smear of each slide. For some strains, a pre-extraction step with formic acid 70%_(v/v)_ or ethanol + formic acid 70%_(v/v)_ + acetonitrile was necessary according to the manufacturer’s instructions. After matrix crystallization, the slides were introduced into the respective piece of equipment. The analysis of the results was in accordance with each manufacturer’s instructions and within the confidence intervals reported above. With VITEK^®^ MS, we used the Saramis Premium (version 4.0.0.14) program. With MALDI Biotyper, we used Biotyper^®^ 2.0 (MBT Compass, version 4.1.100). The strains not identified to the species level by either MALDI-TOF MS system were subjected to complete 16S rRNA gene sequencing.

### 2.3. 16S rRNA Gene Sequencing and Analysis

The complete sequencing of the 16S rRNA gene was performed using the MicroSEQ™ Full Gene 16S rDNA kit (Thermo Fisher Scientific, Waltham, MA, USA) according to the manufacturer’s instructions. The sequences were obtained using the 3500 Series Genetic Analyzer (Applied Biosystems, Waltham, MA, USA) and were processed using DNA Star LaserGene SeqMan software, version 7.0.0. The identification results were obtained from the website https://www.ezbiocloud.net/ (database update: 23 August 2023; last access: 23 November 2023) [12]. All sequences were deposited at https://www.ncbi.nlm.nih.gov/ (database update: 29 December 2023), and accession numbers are provided in Table 2. Complete 16S rRNA gene sequencing results were considered valid when the identification percentage was ≥96%. When the identification was ≥98.7%, the strain was considered identified at the species level [13].

Phylogenetic analysis was conducted through the alignment of the sequences obtained by sequencing the 16S rRNA gene using BioEdit Sequence Alignment Editor software, version 7.0.5.3 [14]. MEGA 11, software version 11.0.13 [12], was used to construct maximum likelihood trees, employing the Kimura-2 parameter model with branching support based on 1000 bootstrap replicates.

## 3. Results and Discussion

The results of MALDI TOF/MS identification are presented in Table 1. Of the 25 strains analyzed, 13 (52.0%) and 12 (48.0%) were not identified by VITEK^®^ MS and MALDI Biotyper^®^, respectively. VITEK^®^ MS identified five (20.0%) strains, which belonged to the genera *Bacillus*, *Paenibacillus*, and *Lysinibacillus*, to the species level. MALDI Biotyper^®^ presented better performance, identifying 13 (52.0%) strains, which belonged to the same genera cited above, at the species level. Unidentified strains by VITEK^®^MS were identified by MALDI Biotyper^®^ as *B. cohnii* and *B. endophyticus*. Only one strain unidentified by MALDI Biotyper^®^ as was identified as *Ureibacillus* spp. by VITEK^®^ MS (Table 1).

The 15 strains unidentified by at least one of the MALDI TOF/MS methods were submitted to full 16S rRNA gene sequencing, providing fragments ranging from 1401 to 1501 base pairs (bp) that were analyzed in the EzBioCloud Database, and the results are shown in Table 2. Five (33.3%) strains were identified at the species level with only one possibility, whereas two strains showed similarity ≤97.04%, indicating that these strains may be new species. These results were similar to those reported by Costa et al. [6], where after 16S rRNA sequencing, 34.02% of 97 strains could be identified at the species level. Considering the 16S rRNA sequencing results as the true-positive results, the strains unidentified by VITEK^®^MS (n = 13) belonged to the genera *Sutcliffiella*, *Bacillus*, *Metabacillus*, *Paenibacillus*, *Neobacillus*, *Priestia*, and *Sporolactobacillus*. The strains unidentified by MALDI Biotyper^®^ (n = 12) belonged to the genera *Bacillus*, *Ureibacillus*, *Paenibacillus*, *Neobacillus*, *Sporolactobacillus*, and *Priestia*. These results indicate the necessity of the expansion of the databases of these methods to allow for the identification of these strains.

Both VITEK^®^MS and MALDI Biotyper^®^ identified three strains as *B. flexus*, and one strain was identified as *Bacillus* spp. by the former and as *B. siralis* by the latter. MALDI Biotyper^®^ also identified one strain as *B. cohnii* (Table 1). These results demonstrate that the databases of both systems have not been updated regarding the species’ nomenclature, as *B. flexus*, *B. spiralis*, and *B. cohnii* have been reclassified to other genera, and currently have the valid names of *Priestia flexa*, *Robertmurraya spiralis*, and *Sutcliffiella cohnii*, respectively [15]. The Bacillaceae family has undergone several reclassifications in its systematics over time. The new techniques used in polyphasic taxonomy have allowed species previously belonging to the genus *Bacillus* to be relocated to other genera [15,16].

Genus-/species-level identification by molecular methodologies can also be used to extend MALDI-TOF MS databases. Database customization is an advantage for microbiology laboratories that need a quick response, such as pharmaceutical industries [10]. This was observed in the case of the genus *Ureibacillus*, which was identified by VITEK^®^MS due to a previous expansion which had included it in the system’s database [11]. The correct identification was also confirmed by 16S rDNA sequencing analysis, where strain B6444 was identified as two possible species: *U. chungkukjangi* or *U. sinduriensis* (Table 2).

Considering the identified strains and those that were correctly identified by their basonym (the earliest validly published name of a taxon), the identification rates at the genus level were 48.0 and 52.0% for VITEK^®^MS and MALDI Biotyper^®^, respectively. The identification rates obtained are low when compared with those reported by Tekippe and Burnham [17], who analyzed 174 “difficult-to-identify” bacterial strains of clinical origin by using VITEK^®^MS and MALDI Biotyper^®^ and obtained identification rates of 86.30 and 92.30%, respectively. As previously stated, the databases of MALDI-TOF MS were initially designed for the analysis of clinical lineages [10], which could explain this discrepancy between the rates of identification in this study, which evaluated pharmaceutical environmental strains. Nevertheless, these rates are similar to those reported by Costa et al. [6], who analyzed 97 aerobic endospore-forming bacteria isolated from a pharmaceutical environment by using VITEK^®^MS and identified 47.42% of the strains at the genus level.

Thus, considering the current nomenclature, the species identified by VITEK^®^ MS, MALDI Biotyper^®^, and 16S rDNA gene sequencing in this study belonged to ten genera: *Bacillus* (n = 9, 36.0%), *Priestia* (n = 5, 20.0%), *Paenibacillus* (n = 4, 16.0%), *Metabacillus* (n = 1, 4.0%), *Robertmurraya* (n = 1, 4.0%), *Lysinibacillus* (n = 1, 4.0%), *Sutcliffiella* (n = 1, 4.0%), *Sporolactobacillus* (n = 1, 4.0%), *Neobacillus* (n = 1, 4.0%), and *Ureibacillus* (n = 1, 4.0%). Mezian et al. [18] analyzed 120 aerobic bacteria endospore-forming strains from a powdered infant formula facility by MALDI-TOF MS and found the genera *Bacillus*, *Paenibacillus*, and *Lysinibacillus*, with *Bacillus* as the most prevalent. Costa et al. [6] also identified *Bacillus* (35.05%), *Paenibacillus* (26.80%), and *Priestia* (9.28%) as the most prevalent genera among 97 strains isolated from a pharmaceutical facility. These studies corroborate the results observed in the present study.

Orem et al. [19] analyzed the phylogenetic diversity of 192 spore-forming aerobic bacillus strains from Brazilian soils using partial 16S rDNA gene sequencing and found that most strains were allocated to the genera *Bacillus* (n = 169; 88.02%), *Paenibacillus* (n = 11; 5.73%), and *Lysinibacillus* (n = 6; 3.13%). These three genera were also found in the present study (Table 1), reinforcing that soil and raw materials can be the main sources of these bacteria in the pharmaceutical production environment. Strains B6099, B791/15, and B1585/18 were analyzed phylogenetically through the construction of phylogenetic trees to observe the closest species (Figure 1 and Figure 2).

B6099 showed the highest percentage of similarity with the species *S. cohnii* (96.37%) and *S. catenulatus* (96.28%). Although the percentages are still lower than the cutoff point to be considered the same species, the phylogenetic tree was built using the species with the highest similarity % according to the EzBioCloud Database. Tree analysis demonstrated that strain B6099 was in a distinct clade from the clade where the species *S. cohnii* and *S. catenulatus* were grouped, reinforcing the possibility that B6099 is a new species of the genus *Sutcliffiella* (Figure 1). However, further studies including the analysis of the complete genome must still be carried out to confirm this hypothesis.

Strains B791/15 and B1585/18 also presented an identification % below the cutoff point for the same species. Because they present greater similarity with species of the genus *Paenibacillus*, they were included in the same phylogenetic tree as species of this genus. Strain B791/15 was most closely related to the species *P. chitinolyticus* (98.50%), which were clustered in the same clade (Figure 2). On the other hand, strain B1585/18 was most closely related to the species with *P. provencensis* (97.04%) but was allocated to a distinct clade (Figure 2). This result indicates the possibility that strain B1585/18 is a new species within the genus *Paenibacillus*. However, further studies including WGS analysis should be carried out to confirm this hypothesis.

In this study, the most prevalent genera were *Bacillus*, *Priestia*, and *Paenibacillus*. Microorganisms belonging to the genera *Bacillus* and *Paenibacillus* are characterized as large Gram-positive bacilli, which form spores resistant to heat, cold, and common disinfectants, allowing them to survive on environmental surfaces for prolonged periods. They are frequently found in soil, air, water, and food [8]. The genus *Priestia* encompasses aerobic endospore-forming rods that are mainly Gram-positive, but there are some species that present Gram-negative (*P. koreensis*) and Gram-variable (*P. flexa*) staining. They can be isolated from various sources, including feces, soil, internal tissues of cotton plants, marine sediments, and the rhizosphere of willow roots [15]. The primary niches of these genera make them potential contaminants of pharmaceutical production environments.

Among the 13 strains identified at the genus level, 10 (40.0%) presented >98.7% of similarity with more than one species of the genus (Table 2). Studies using molecular methods showed that species boundaries between members of *Bacillus* and related genera are difficult to define, requiring analysis of other housekeeping genes, such as *rpoB* or *gyrB* [18,20].

Strain B773/19 presented *B. pseudoflexus* as a possibility, with 99.65% of similarity (Table 2). *B. pseudoflexus* was described by Chandna et al. [21] from a compost sample. At the time of analysis, the 16S rRNA gene analysis of *B. pseudoflexus* RC1^T^ FN999944 clustered with strains *B. paraflexus* MTCC 9831^T^ FN999943 and *B. flexus* DSM 1320^T^ ABO21185. These species have been reclassified into the genus *Priestia* as *P. paraflexa* and *P. flexa*, respectively [15]. In addition, the *B. pseudoflexus* 16S rRNA gene sequence was more closely related to the species *Priestia* than to *Bacillus* (Figure 3). These data support the hypothesis that strain *B. pseudoflexus* RC1^T^ FN999944 is representative of a novel species in the genus *Priestia*. However, the species *B. pseudoflexus* is not included in the List of Prokaryotic names with Standing in Nomenclature (https://lpsn.dsmz.de/search?word=bacillus; accessed on 1 November 2023) [22]. This is because the complete genome sequenced of this species is not available yet, so the species *B. pseudoflexus* has not been validly published [15]. Consequently, the reclassification of *B. pseudoflexus* to the genus *Priestia* as *Priestia pseudoflexus* sp. nov. is proposed.

### Description of Priestia pseudoflexus sp. nov.

Pries’ti.a. N.L. fem. n. Priestia named for British microbiologist Prof. Fergus G. Priest (Heriot-Watt University, Edinburgh; 1948–2019) for his many contributions to the systematics and uses of the members of the genus *Bacillus*.

Pseudoflexus: pseu.do.fle’xus. Gr. adj. pseudês, false; L. masc. adj. flexus, flexible and also a bacterial epithet [B. flexus (ex Batchelor 1919); [23]]; N.L. masc. adj. pseudoflexus, the false [Bacillus] flexus, of compost.

Basonym: Bacillus pseudoflexus Chandna et al. [21].

The description of this species is the same as that provided by Chandna et al. [21] for *Bacillus pseudoflexus*.

The type strain, RC1T (=MTCC 9830T=KCTC 13723T=CCM 7753T), was isolated from a compost in India. The genomic DNA G+C content of the type strain is 40.4 ± 0.2 mol%.

## 4. Conclusions

The MALDI-TOF MS methodology was not able to identify most spore-forming aerobic bacteria isolated from a pharmaceutical industry facility to the species level. Therefore, there is a need for the expansion of the database of these methods to allow for the identification of these strains.

16S rRNA gene sequencing was insufficient to identify species boundaries between members of *Bacillus* and related genera and requires the sequencing of other housekeeping genes, such as *rpoB* or *gyrB*. Therefore, further analysis will be necessary to try to identify ten strains in this study at the species level.

Other studies in the literature have shown that since the environment of the pharmaceutical industry is little studied, there is room for the discovery of new bacterial species. For these species to be described, further taxonomic analyses will be necessary.

Furthermore, as there is no possibility of analyzing the complete genome of *B. pseudoflexus*, we propose the reclassification of this species to the genus *Priestia* using the total sequencing of the 16S rRNA gene, since this species is grouped in the same clade as other strains of the genus *Priestia*.

The need for identification of strains isolated from pharmaceutical production environments at the species level demands a search for fast, accurate methodologies that are reproducible and low-cost. Each methodology has its own limitations; thus, one tool alone will not always be sufficient to identify a bacterial strain at the species level. Therefore, polyphasic taxonomy, with the use of various phenotypic and genotypic methodologies, is necessary to reach the lowest taxonomic level of identification.

## Figures and Tables

**Figure 1 microorganisms-12-00724-f001:**
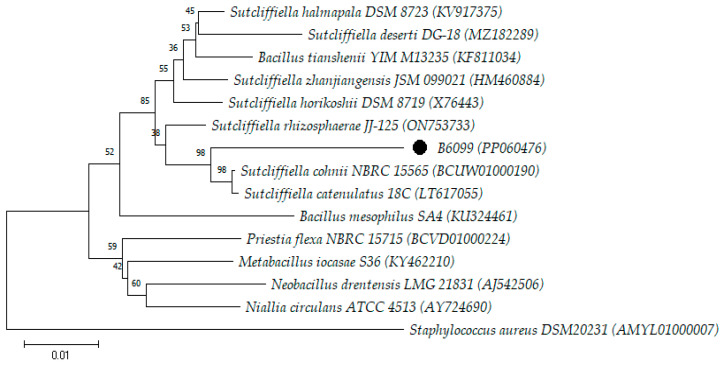
Neighbor-joining tree based on almost complete sequences of the 16S rRNA gene (1374 bp) showing the phylogenetic position of isolate B6099 with the closest species of the genera *Sutcliffiella*, *Bacillus*, *Neobacillus*, and *Niallia*. *Staphylococcus aureus* DSM 20231 was used as an outgroup. The numbers at the nodes indicate the percentage of 1000 bootstrap replicates. The scale bar represents 0.01 substitutions per nucleotide position. The GenBank accession numbers are provided in parentheses.

**Figure 2 microorganisms-12-00724-f002:**
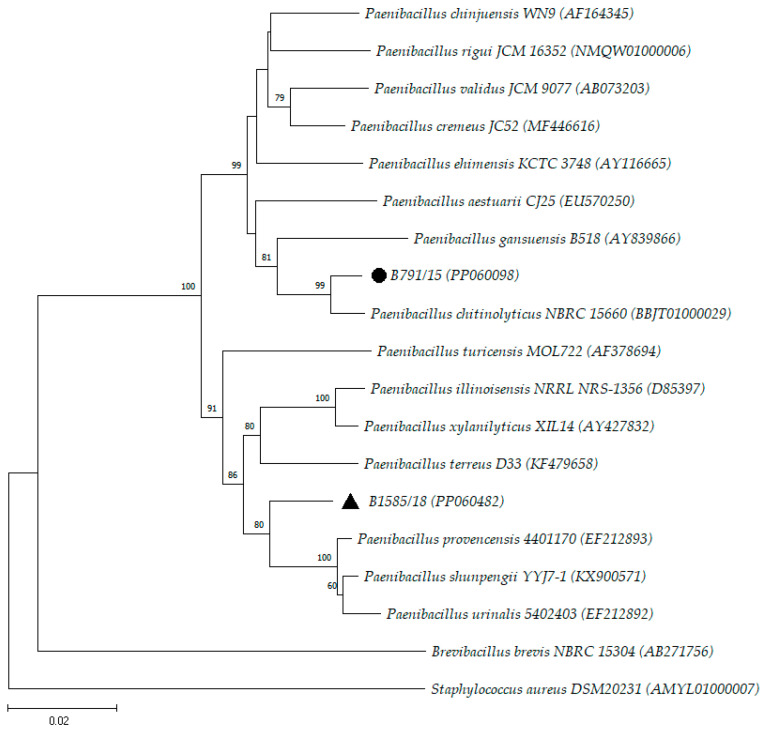
Neighbor-joining tree based on almost complete sequences of the 16S rRNA gene (1323 bp) showing the phylogenetic position of strains B791/15 (●) and B1585/18 (▲) with the closest species of the genus *Paenibacillus*. *Brevibacillus brevis* NBRC 15304 and *Staphylococcus aureus* DSM 20231 were used as outgroups. The numbers at the nodes indicate the percentage of 1000 bootstrap replicates. The scale bar represents 0.02 substitutions per nucleotide position. The GenBank accession numbers are provided in parentheses.

**Figure 3 microorganisms-12-00724-f003:**
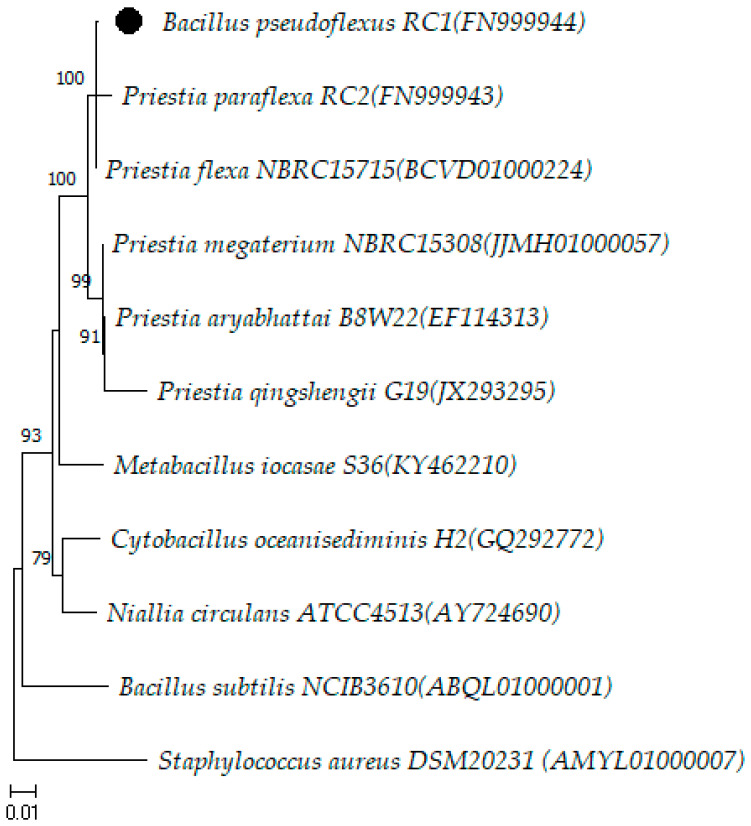
Neighbor-joining tree based on almost complete sequences of the 16S rRNA gene (1351 bp) showing the phylogenetic position of *Bacillus pseudoflexus* (●) with the closest species of the genus *Bacillus* and related genera. *Staphylococcus aureus* DSM 20231 was used as outgroup. The numbers at the nodes indicate the percentage of 1000 bootstrap replicates. The scale bar represents 0.01 substitutions per nucleotide position. The GenBank accession numbers are provided in parentheses.

**Table 1 microorganisms-12-00724-t001:** Results of aerobic endospore-forming bacteria (n = 25) identification by VITEK^®^ MS and MALDI Biotyper^®^.

Strain	Origin	VITEK^®^ MS (%)	MALDI Biotyper^®^ (Score)
B6081	Air monitoring	*Paenibacillus* spp. (83.80)	*P. glucanolyticus* (1.88)
B6099	Water vapor	NI	*B. cohnii* (1.94)
B6114	Water vapor	NI	NI
B6389	Operator monitoring	NI	NI
B6444	Bioburden test	*Ureibacillus* spp. (84.60)	NI
B791/15	Potable water	NI	NI
B1513/15	Sterility test	NI	NI
B636/16	Sterility test	NI	NI
B669/18	Operator monitoring	NI	NI
B1439/18	Bioburden test	NI	*B. endophyticus* (1.74)
B1567/18	Potable water	NI	NI
B1585/18	Air monitoring	NI	NI
B554/19	Bioburden test	NI	NI
B773/19	Bioburden test	NI	NI
B1124/19	Bioburden test	NI	NI
B1665/18	Bioburden test	*P. validus* (99.90)	*P. validus* (2.21)
B1710/18	Bioburden test	*B. licheniformis* (99.90)	*B. licheniformis* (2.14)
B1711/18	Bioburden test	*Bacillus* spp. (81.20)	*B. siralis* (2.15)
B1819/18	Bioburden test	*B. cereus* group (99.90)	*B. cereus* (2.09)
B1847/18	Bioburden test	*B. cereus* group (99.90)	*B. cereus* (2.15)
B1972/18	Bioburden test	*B. cereus* group (89.90)	*B. cereus* (2.14)
B1192/19	Purified water	*L. fusiformis*/*sphaericus* (99.00)	*L. fusiformis* (2.06)
B1209/19	Bioburden test	*B. flexus* (84.60)	*B. flexus* (2.26)
B1213/19	Sterility test	*B. flexus* (75.60)	*B. flexus* (2.23)
B097/20	Bioburden test	*B. flexus* (88.20)	*B. flexus* (2.36)

NI—not identified.

**Table 2 microorganisms-12-00724-t002:** Identification by 16S rRNA gene sequencing of the strains (n = 15) not identified by either MALDI TOF/MS system.

Strain	NCBI ¹ Access Number	bp ²	Species (% of Similarity)
B6081	PP059127	1501	*P. lautus* (99.25)
B791/15	PP060098	1401	*P. chitinolyticus* (98.50)
B1585/18	PP060482	1488	*P. provencensis* (97.04)
B6099	PP060476	1447	*S. cohnii* (96.37)
B6389	PP059202	1426	*M. idriensis* (99.35)
B554/19	PP060516	1470	*S. inulinus* (98.83)/*S. terrae* (98.77)
B636/16	PP060583	1432	*N. drentensis* (99.42)/*N. soli* (99.16)/*N. bataviensis* (99.02)/*N. vireti* (99.01)/*N. cucumis* (98.95)/*N. novalis* (98.81)
B1439/18	PP060597	1495	*P. filamentosa* (99.86)/*P. endophytica* (99.32)
B773/19	PP060628	1457	*P. flexa* (99.86)/*B. pseudoflexus* (99.65)/*P. paraflexa* (98.84)/*P. megaterium* (98.83)/*P. aryabhattai* (98.69)
B6444	PP060737	1462	*U. chungkukjangi* (99.86)/*U. sinduriensis* (99.00)
B6114	PP060738	1445	*B. paralicheniformis* (99.79)/*B. glycinifermentans* (99.58)/*B. haynesii* (99.51)/*B. sonorensis* (99.24)/*B. licheniformis* (99.17)/*B. aerius* (98.75)
B1513/15	PP060747	1443	*B. siamensis* (99.93)/*B. velezensis* (99.93)/*B. subtilis* (99.79)/*B. amyloliquefaciens* (99.72)/*B. nakamurai* (99.65)/*B. tequilensis* (99.58)/*B. cabrialesii* (99.58)/*B. inaquosorum* (99.58)/*B. stercoris* (99.58)/*B. vallismortis* (99.51)/*B. atrophaeus* (99.44)/*B. halotolerans* (99.44)/*B. spizizenii* (99.44)/*B. mojavensis* (99.37)
B669/18	PP060782	1492	*B.velezensis* (99.93)/*B. siamensis* (99.86)/*B. subtilis* (99.59)/*B. amyloliquefaciens* (99.59)/*B. nakamurai* (99.59)/*B. atrophaeus* (99.39)/*B. tequilensis* (99.39)/*B. halotolerans* (99.39)/*B. cabrialesii* (99.39)/*B. inaquosorum* (99.39)/*B. stercoris* (99.39)/*B. mojavensis* (99.32)/*B. vallismortis* (99.32)/*B. spizizenii* (99.25)
B1567/18	PP060923	1465	*B. paralicheniformis* (99.66)/*B. glycinifermentans* (99.45)/*B. haynesii* (99.38)/*B. licheniformis* (99.11)/*B. sonorensis* (99.11)/*B. aerius* (98.70)
B1124/19	PP060925	1480	*B. licheniformis* (99.52)/*B.haynesii* (99.45)/*B. sonorensis* (99.32)/*B. paralicheniformis* (99.18)/*B. aerius* (99.11)/*B. glycinifermentans* (98.98)/*B. swezeyi* (98.77)

¹ National Center for Biotechnology Information; ^2^ base pair length.

## Data Availability

Data are contained within the article.

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
