# Peer review of "Characterization by MALDI-TOF MS and 16S rRNA Gene Sequencing of Aerobic Endospore-Forming Bacteria Isolated from Pharmaceutical Facility in Rio de Janeiro, Brazil"

_microorganisms, 2024, doi:10.3390/microorganisms12040724_

Round 1

Reviewer 1 Report

Comments and Suggestions for Authors

This study identified aerobic endospore-forming bacteria isolated from a pharmaceutical facility by MALDI-TOF MS and 16S rRNA gene sequencing. Although this work is meaningful in this field, the manuscript is not reach the level of acceptance for publication at current version.

Main concerns:

(1) To my best knowledge, the MALDI-TOF MS and 16S rRNA gene sequencing could only identified strain at genus level. It is not very accurate to identified strain at species level by MALDI-TOF MS and 16S rRNA gene sequencing.

(2) The reclassification of Bacillus pseudoflexus to the genus Priestia is also need more proofs such as ANI anlysis based on genome, traditional biochemical experiment, types of Fatty acids, respiratory quinones, etc.

Overall, the results can not fully support the insight of this manuscript.

Reviewer 2 Report

Comments and Suggestions for Authors

The aim of this research was to identifiy microorganisms at the species level by 16S rDNA and MALDI TOF/MS, that could be eventually published if it will be extensively revised.

To improve the manuscript, I would suggest the following:

1) L26-27, ‘16S rDNA and MALDI TOF/MS were not sufficient to identify all strains until species level, and complementary analysis are necessary’.

So, why authors use those tools, and do not provide the complementary analysis?

2) Authors should provide the advantages and disadvantages of 16S rDNA and MALDI TOF/MS in the section of introduction, so authors combine the two.

3) What is the difference between MALDI Biotyper® and VITEK® MS RUO systems?

4) L52-54, ‘MALDI-TOF MS technology, has proven to be a good option for microorganisms’ identification from pharmaceutical origin’.

Could this method been applied in other industries?

5) L125-126, ‘MALDI Biotyper® presented a better performance, identifying 13 (52.0 %) strains to the species level’.

Why did authors not use 16S rDNA to verify the 13 strains?

And why did MALDI Biotyper® have a better performance?

6) Figures should be uniformed.

7) Please add conclusion or discussion.

8) L27-30, ‘As new possible species were identified, the complete genome sequencing is necessary to evaluate and describe them. The expansion of the MALDI- TOF MS database with these species can make these identification processes faster, improving quality control in the pharmaceutical industry.’

It is just the outlook of MALDI- TOF MS for identification of species, which is not appropriate in the section of abstract.

Authors should revise the whole abstract.

Comments on the Quality of English Language

Moderate editing of English language required.

Reviewer 3 Report

Comments and Suggestions for Authors

The authors identified several strains as potential contaminants of a pharmaceutical facility by MALDI-TOF MS and 16S rRNA gene sequencing. I agree with the authors that this research is of value for future pharmaceutical production. However, some flaws can be found in the manuscript, especially for the technologies the used for identification of microbes.

Major comments

1. MALDI-TOF MS and 16S rRNA gene sequencing are indeed the effective methods for bacterial identification. But I do not think these methods are convincing for the identification of novel species without any information of biochemical indicator analysis and genome sequencing. Therefore, I really doubt the proposal of Priestia pseudoflexus sp. nov. in the result.

2. The authors state that the aim of this study was to compare three methodologies for identification of endo-spore-forming aerobic bacteria strains. However, I cannot find the result supported the comparison. Which one is better or accuracy for the identification? The authors should organize the manuscript around their aim. Now, the manuscript mainly describes the microbial identification instead of the related methods. The current result seems to say that 16S rRNA gene sequencing is a supplemental method for MALDI-TOF MS.

3. The manuscript emphasizes the bacterial taxonomic information helps improving the quality control in the pharmaceutical industry. At least, the authors should offer the readers a picture that how the taxonomic information in their work promotes the pharmaceutical industry?

Minor comments

1. Line 74, the authors should provide sampling process, isolation method, and medium components for the bacterial isolation or related references.

2. Genus name Bacillus should be written in abbreviation B. when it appears again in the manuscript.

3. The figures of phylogenetic trees are too blurred for a clear observation.

4. How the authors confirm these strain are aerobic endospore-forming bacteria? The supporting results have to show in the manuscript.

Reviewer 4 Report

Comments and Suggestions for Authors

Dear Authors

The text of the manuscript submitted for review seems interesting and concerns the problem of product contamination during the preparation process.

However, the manuscript requires proofreading to improve its readability, understanding and editability.

It is also necessary to introduce a Conclusions chapter.

The authors should also add more literature, especially regarding comparative analysis of the effectiveness of the methods used in the research.

The authors should also answer the main question: is it necessary to use so many research methods at the same time to identify bacteria?

All necessary comments are included in the text of the manuscript.

In it's current form, the manuscript requires minor revision.

Reviewer 5 Report

Comments and Suggestions for Authors

This work reports work identifying bacterial species contaminating a pharmaceutical facility, focussing on aerobic spore formers. The work appears well done, and the conclusions justified. However there are a number of significant concerns that need to be addressed. These include the following.

1) l 24 - not clear why this high degree of sequence similarity indicates these are a new species.

2) l 88 "of the "What" was applied" - and how was this obtained.

3) l 88-93 Overall, I found the description of the sample prep for proteomic analysis missing important information such as how proteins were extracted, was there proteolysis, etc, and also the analyses of the proteomic data.

3) l 96 - similarity to what, and seems like a reference is needed here.

4) l 134/135 - not clear why two strains having high similarity are likely a new species.

5) Table 1  - (Species(% of similarity) - similarity to what?

6) Were there non-spore forming organisms in phamceutical plants>

Comments on the Quality of English Language

The English in this manuscript is certainly understandable, but will need some editing to smooth out many cases of incorrect English usage.

Examples include (and also many more): i) l 35/36 - "...risk to product safety..."; 2) l 67 - delete "the"; 3) l 89 delete microliter; 4) l 126 - "One strain not identified...; 5) l 131 - "The 15 strains unidentified...; 6) l 136 - delete "possibly, and change "that" to "those"; 7) l 77 - "...by the Vitek..."

Round 2

Reviewer 2 Report

Comments and Suggestions for Authors

The manuscript has been modified. Nevertheless, there're a series of problems:

1. It is a good option for microorganisms’ identification by MALDI-TOF MS systems, but molecular methodologies are irreplaceable for validation of new methods.

So additional experiments needed.

2. ‘Discussions can be combined with Results’, but there is lack of summary for manuscript in the section of Results.

3. Figures should be uniformed.

Reviewer 3 Report

Comments and Suggestions for Authors

Thank you for all the responses.
